# Benefits of Inactivated Vaccine and Viral Vector Vaccine Immunization on COVID-19 Infection in Kidney Transplant Recipients

**DOI:** 10.3390/vaccines10040572

**Published:** 2022-04-08

**Authors:** Napun Sutharattanapong, Sansanee Thotsiri, Surasak Kantachuvesiri, Punlop Wiwattanathum

**Affiliations:** 1Division of Nephrology, Department of Medicine, Faculty of Medicine Ramathibodi Hospital, Mahidol University, 270 Rama 6 Road, Ratchathewi, Bangkok 10400, Thailand; napun.sut@mahidol.ac.th (N.S.); sansanee.tht@mahidol.ac.th (S.T.); surasak.kan@mahidol.ac.th (S.K.); 2Excellent Center for Organ Transplantation, Faculty of Medicine Ramathibodi Hospital, Mahidol University, Bangkok 10400, Thailand

**Keywords:** inactivated vaccine, viral vector vaccine, COVID-19, kidney transplantation

## Abstract

The coronavirus virus disease 2019 (COVID-19) pandemic has impacted the global healthcare system. In Thailand, the first and most available vaccines were inactivated and viral vector vaccines. We reported the impact of those vaccines in preventing severe disease and death in kidney transplant recipients. This retrospective study comprised 45 kidney transplant recipients with COVID-19 infection, classified by vaccination status. Outcomes of interest were death, pneumonia, and allograft dysfunction. There were 23 patients in vaccinated group and 22 patients in unvaccinated group. All baseline characteristics were similar except mean age was older in vaccinated group, 55 vs. 48 years. Total 11 patients (24%) died (13% vaccinated vs. 36% unvaccinated RR, 0.56; 95% CI, 0.29–0.83; *p* = 0.03). Multivariate analysis showed that vaccination significantly decrease mortality (odds ratio, 0.54; 95% CI, 0.10–0.94; *p* = 0.03). Pneumonia developed equally in both groups (70%). There was a trend toward less oxygen requirement as well as ventilator requirement in vaccinated group. The rate of allograft dysfunction was similar (47%). Inactivated and viral vector COVID-19 vaccines have beneficial effect on mortality reduction in kidney transplant recipients. Even partial vaccination can exert some protection against death. However, full vaccination should be encouraged to achieve better prevention.

## 1. Introduction

The novel coronavirus causing severe acute respiratory distress syndrome (COVID-19) pandemic has impacted the global healthcare system. According to the burden of the disease and the lack of effective treatments, the strategy was focused on vaccine development. Recent studies in general population showed the efficacy of vaccination in preventing COVID-19 infection and also a reduction in severity and mortality, by inducing humoral and cellular immune response [1,2,3,4,5,6]. Immune response after vaccination in transplant recipients is markedly lower than immunocompetent counterpart [7]. Studies of humoral immune response after variety of vaccination platform showed negligible antibody seroconversion after inactivated vaccine [8]. Studies on viral vector and mRNA vaccine showed higher seroconversion after mRNA vaccine than after viral vector vaccine. On the other hand, cell-mediated immune response can be achieved after all types of vaccine [9,10,11,12,13]. The above-mentioned studies did not report clinical outcome. In addition to lower immune response after vaccination, the severity and mortality rate are much higher in unvaccinated kidney transplant recipients than general population (19–38% vs. 2–3%) [14,15,16,17]. Messenger ribonucleic acid (mRNA) vaccine have shown to reduce severity and mortality significantly in kidney transplant recipients [18]. There were scant reports on the efficacy of inactivated vaccine or viral vector vaccine in transplant recipients. Initially, mRNA vaccines were available in developed countries, but the first and most available vaccines in Thailand were inactivated and later on, viral vector vaccine. When vaccines became available, the amount were limited so prioritization was introduced to vaccinate people who were believed to be at high risk for developing severe disease and thus received highest benefit from vaccination. Unfortunately, kidney transplant recipients did not receive priority and vaccine hesitancy was prevalent among some patients [19]. With above reasons, only minority of kidney transplant recipients received vaccination during the study period. This study aims to demonstrate the impact of inactivated and viral vector vaccines on severity and mortality reduction in COVID-19 infected kidney transplant recipients.

## 2. Methods

### 2.1. Study Design and Participants

This is a retrospective cohort study of kidney transplant patients at Ramathibodi Excellent Center for Organ Transplantation who were diagnosed with COVID-19 infection between January 2021 and October 2021. All adult recipients, aged 18 years or older, who received either living or deceased donor kidney transplantation were eligible for the study. Patients with symptoms suggestive of having COVID-19 infection or asymptomatic patients who had contacted with COVID-19 cases were tested. Diagnosis was confirmed by real-time polymerase chain reaction (RT-PCR) for severe acute respiratory syndrome coronavirus 2 (SARS-CoV2). The study protocol was approved by the Institutional Human Research Ethics Committee of Faculty of Medicine Ramathibodi Hospital, Mahidol University [MURA 2021/895]. The patients were categorized by vaccination status. COVID-19 vaccines available in Thailand at the time of this study were CoronaVac inactivated vaccine (Sinovac Biotech) and ChAdOx1 nCoV-19 adenoviral vector vaccine (Vaxzevria, Oxford-AstraZeneca). Inactivated vaccines were given 4 weeks apart [20] while viral vector vaccines were given 12 weeks apart [21]. Patients who received 2 doses of either vaccine were considered fully vaccinated and those who received only 1 dose were considered partially vaccinated.

### 2.2. Outcomes

The primary objective of this study was to assess the impact of inactivated and viral vector vaccines in mortality reduction in COVID-19 infected kidney transplant recipients. The observation endpoint was defined when the patients were discharged from the hospital, or hotel-based self-care program. Secondary outcomes included severity of COVID-19 disease, complication, oxygen and ventilator support requirement, and acute kidney injury (AKI) during the hospital course. AKI was defined and staged by using Kidney Disease: Improving Global Outcomes (KDIGO) [22].

### 2.3. Management during COVID-19 Infection

According to the local standard of care for COVID-19 infection, patients have been classified by the severity and risk factors of severe disease. Severe cases were admitted to the hospital and those who had mild symptoms or asymptomatic were assigned to hotel-based self-care. Kidney transplant recipients were always classified as high risk of severe disease due to immunosuppressive state, thus they all received antiviral agent, favipiravir, as soon as the RT-PCR was confirmed. If the patients required oxygen therapy, they were prescribed dexamethasone 6–20 mg daily according to the treating physicians. Immunomodulatory agents or hemoperfusion was considered in case of severe pneumonia. The immunosuppressive medication was adjusted according to the severity of disease. If the patients develop any symptoms, anti-proliferative agents were discontinued and calcineurin inhibitors (CNIs) were reduced toward the low therapeutic level. Corticosteroids were continued or resumed if steroid-free regimens were used. If there is an indication for methylprednisolone or dexamethasone, prednisolone was substituted. 

### 2.4. Statistical Analysis

Baseline characteristics were described as frequency (percent) for categorical data. Chi-square test was used for comparison between groups. Continuous measurements were reported as mean (standard deviation) for normal, and median (interquartile range) for non-normal distribution. Depending on data distribution, independent *t*-test and Mann-Whitney test were used to compare their differences [23]. 

The primary outcome was mortality and evaluated by logistic regression analysis as relative risk ratio (RR) of death between groups with 95% confident intervals (CIs), and also reported in Kaplan-Meier time-to-event analysis using log-rank test [24]. We also reported secondary outcomes as a relative risk ratio with 95% CIs. Univariate and multivariate logistic regression analyses were used to identify factors that may affect mortality (vaccination status, recipient age, type of donor, diabetes, obesity and sepsis). All statistical analyses were performed with IBM SPSS software, version 23. 

## 3. Results

### 3.1. Patients

During the study period, there were 45 kidney transplant recipients with confirmed COVID-19 infection. Twenty-three patients (51%) received COVID-19 vaccine. The median post-transplant time was 5.4 years and mean baseline renal function assessed by estimated glomerular filtration rate (eGFR) was 62.1 mL/min/1.73 m^2^. The baseline characteristics were shown in Table 1.

Of the 23 patients, 18 (71%) received ChAdOx1 nCoV-19 vaccine, of which only 2 patients were fully vaccinated with 2 doses. The remaining 5 patients (29%) were fully vaccinated with 2 doses of CoronaVac vaccine. Infection was diagnosed with median of 49 days after the last dose of vaccination (mean of 37 days in patients who received only one dose of vaccine and 71 days in those who received two doses). The mean age of patients in vaccinated group was higher than unvaccinated group, 55 vs. 48 years, respectively (*p* = 0.04).

### 3.2. Hospital Course

Ninety-five percent of COVID-19 infection in kidney transplant recipients were symptomatic. Pneumonia was found in 70% of patients in both groups (*p* = 0.89). Gastrointestinal symptoms were found in only 5% of the patients. Twelve patients (52%) in vaccinated group required oxygen therapy, while 16 patients (71%) in unvaccinated group needed oxygen supplement (*p* = 0.10). More patients in unvaccinated group required mechanical ventilator support (19% vs. 9%; *p* = 0.48). All patients received favipiravir as antiviral agent for 5–10 days and nearly half required corticosteroids (either dexamethasone or methylprednisolone) for the treatment of COVID-19. Biologic immunomodulatory agents or hemoperfusion were required in a few patients. Immunosuppressive management was the same in both groups, i.e., continue CNIs, discontinue mycophenolate in the majority of the patients (Table 2).

### 3.3. Clinical Outcomes

The mortality rate was 24.4% overall and significantly lower in vaccinated group than unvaccinated group (Figure 1), 13% vs. 36% (RR, 0.56; 95% CI, 0.29–0.83; *p* = 0.03). There was no mortality in patients who received 2 doses of vaccine, either inactivated (*n* = 5) or viral vector (*n* = 2) vaccine. Even with partial vaccination (only 1 dose of vaccine), there was a trend toward mortality reduction compared with no vaccination, 19% and 36%, respectively (*p* = 0.08). There was no significant difference in other complications such as bacterial pneumonia, invasive pulmonary aspergillosis, pulmonary embolism, or septicemia (Table 3). 

Univariate analysis revealed that diabetes and vaccination affected mortality. Multivariate analysis showed that diabetes significantly increase mortality (odds ratio, 5.37; 95% CI, 1.07–26.95; *p* = 0.04) whereas vaccination decrease mortality (odds ratio, 0.54; 95% CI, 0.10–0.94; *p* = 0.03, Table 4). The incidence of allograft dysfunction in COVID-19 infected kidney transplant recipients was 47%, similar between groups (*p* = 0.64). Most allograft dysfunctions were Acute Kidney Injury Network (AKIN) stage 1, caused by prerenal which improved after intravenous hydration. Only one patient in each group (4%) requires intermittent hemodialysis for kidney replacement therapy.

## 4. Discussion

Our study showed overall mortality of COVID-19 infection of 24.4% in kidney transplant recipients, similar to previous reports ranging from 18–37% [14,15,16,17]. Kidney transplant recipients had a much greater mortality rate than the overall population, especially those with diabetes. Studies have shown correlation between age and severity of disease, i.e., the higher the age of the patients, the higher the mortality [25,26]. In our study, the vaccinated group was older than unvaccinated group but lower mortality rate. After adjusting with multivariate analysis, this study demonstrated that inactivated and viral vector vaccines can significantly reduce mortality in COVID-19 infected kidney transplant recipients. In our study, no patients who received 2 doses of vaccine succumbed to the disease and patients who received only one dose of vaccine had mortality rate of 19% which was lower than 36% in unvaccinated group albeit not statistically significant. Our data suggested that kidney transplant recipients should be fully vaccinated as soon as possible to acquire the full benefit. There were reports showing that longer interval (12 weeks) between first and second dose of viral vector vaccine produced better immune response [6,27]. Our study showed that the mean interval from vaccination in patients who received only one dose of vaccine to the time of confirmed diagnosis of COVID-19 infection was 37 days. In situation of infection outbreak, shorten the interval between the 2 doses may be a reasonable practice to provide better protection for the patients. Another strategy is to use vaccine platform with short interval between doses such as inactivated or mRNA vaccine.

Management of COVID-19 infection in kidney transplant recipients was similar to that for general population except antiviral agent. They all received antiviral agent, favipiravir, due to their immunosuppressive state and high risk for disease progression. The intensity of corticosteroids was adjusted according to the oxygen therapy requirement. In our study, 45% required dexamethasone or methylprednisolone. At Ramathibodi Hospital, we manage immunosuppressive medication following the ERA-EDTA DESCARTES expert opinion [28], which is similar to other reports [15,29,30]. For antimetabolite, mycophenolate was mostly withheld during COVID-19 infection and resumed 1–2 weeks after infection resolved. CNIs were continued the same dose as prior to infection. Only 7% of the cohort had to withdraw CNIs due to severe diseases i.e., requirement of mechanical ventilation, hemodynamic instability, or complicated with opportunistic infection. Thus, the rate of allograft rejection after COVID-19 infection is a challenging issue in managing this group of patients.

Not only the mortality, the COVID-19 infected kidney transplant recipients developed pneumonia more often than the general population. Our study showed the incidence of pneumonia in kidney transplant recipients at 70%, while Pongpirul et al. [31] reported 39% in general population. Inactivated and viral vector vaccines did not reduce the incidence of pneumonia which may be from a very low number of fully vaccinated patients, but there was a trend toward less oxygen requirement in vaccinated patients (52% vs. 71%, *p* = 0.10) as well as less ventilator requirement (9% vs. 19%, *p* = 0.48). This implied that vaccination could reduce the severity of the disease. Meshram et al. [32] reported 4 kidney transplant recipients developing COVID-19 infection after 2 doses of ChAdOx1 nCoV-19, viral vector vaccines. One patient died due to acute respiratory distress syndrome (ARDS) and 2 patients required mechanical ventilation. Our study demonstrated the benefit of inactivated and viral vector vaccines in terms of decreased mortality and severity. 

Recent studies showed a suboptimal humoral immune response to mRNA COVID-19 vaccine among kidney transplant recipients measured by anti-spike antibody [9,10,11,13,33]. Study by Bruminhent et al. [8] on the effect of immune response in patients receiving inactivated vaccine showed that humoral immune response was almost negligible after 2 doses of vaccine. On the other hand, cell mediated immune response could be achieved at the level almost the same as normal population. For viral vector vaccine, cell mediated immune response could be measured to be stimulated after just one dose of vaccine but humoral immune response was poor [34]. Recent data showed a good efficacy of the third dose of mRNA vaccine which improved humoral immune response in kidney transplant recipients from 50% to 70% [35]. However, the efficacy of booster dose with mRNA vaccine after other forms (inactivated or viral vector) of COVID-19 vaccines is unknown and required further study. Although in real life, we did not measure the immune response in our patients, it was assumed that the response would be the same and there was benefit of vaccination in reducing mortality significantly. This underscores the importance of cellular immune response in fighting viral infection.

Renal allograft dysfunction occurred in 47% of cases. The rate of allograft dysfunction did not differ between groups and only 10% of those required renal replacement therapy. The incidences were similar to the previous studies which reported ranging 40–75% and 6.6–46%, respectively [14,15].

The limitation of this study is the small number of vaccinated subjects included in the study. In Thailand, due to vaccine shortages during the study period, vaccination initially prioritize given to population 60 years or older and those with some specified comorbidity. Transplant recipients did not receive priority for vaccination. This may limit the number of kidney transplant recipients receiving vaccination and make a demonstration of vaccine efficacy in reducing severity other than death difficult due to a small number of vaccinated transplant recipients.

## 5. Conclusions

The mortality and severity of COVID-19 infection among kidney transplant recipients were significantly higher than the general population. Inactivated and viral vector COVID-19 vaccines have beneficial effect on mortality reduction in kidney transplant recipients. Even partial vaccination can exert some protection against death. However, to achieve better prevention, full vaccination should be encouraged for all kidney transplant recipients.

## Figures and Tables

**Figure 1 vaccines-10-00572-f001:**
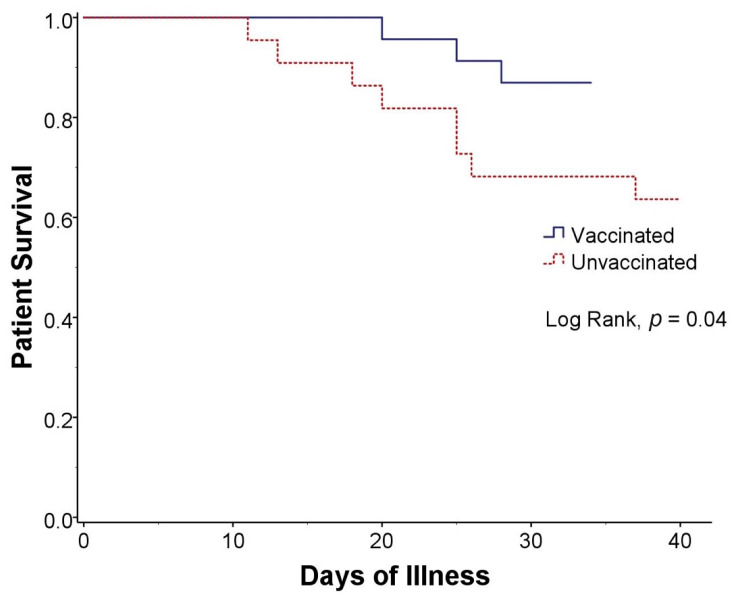
Kaplan-Meier curve of survival in kidney transplant recipients with COVID-19 infection who were vaccinated (blue solid line) and unvaccinated (red dashed line).

**Table 1 vaccines-10-00572-t001:** Baseline characteristics.

Patient Characteristics	Total (N = 45)	Vaccinated (N = 23)	Unvaccinated (N = 22)	*p*-Value
**Age (y), mean** **(** **SD)**	52 (13)	55 (11)	48 (15)	0.04
**Gender, *n* (%)**				0.12
Male	30 (67)	18 (78)	12 (55)	
Female	15 (33)	5 (22)	10 (45)	
**BMI (kg/m^2^), mean (SD)**	25.11 (5.68)	25.19 (5.49)	25.03 (6.00)	0.92
**Kidney transplant status**				0.67
Deceased donor transplantation, *n* (**%**)	30 (67)	16 (70)	14 (64)	
Living related transplantation, *n* (**%**)	15 (33)	7 (30)	8 (36)	
Post-transplant time (y), median (IQR)	5.4 (2.63–8.65)	4.6 (2.74–7.69)	6.72 (2.28–9.66)	0.86
Baseline creatinine (mg/dL), mean (SD)	1.45 (0.67)	1.46 (0.52)	1.43 (0.81)	0.85
eGFR (mL/min/1.73 m^2^), mean (SD)	62.1 (26.5)	60.6 (26.3)	63.7 (27.2)	0.69
**Immunosuppressive regimens, *n*** **(** **%** **)**				
Calcineurin inhibitors	41 (91)	22 (96)	19 (86)	0.35
Mycophenolate	42 (93)	23 (100)	19 (86)	0.11
mTOR inhibitors	3 (7)	1 (4)	2 (9)	0.61
Steroids	43 (96)	21 (91)	22 (100)	0.49
**Comorbidity, *n* (** **%** **)**				
Diabetes	20 (44)	10 (43)	10 (45)	0.89
Hypertension	32 (71)	18 (78)	14 (64)	0.34
Ischemic heart disease	7 (16)	5 (22)	2 (9)	0.41
Obesity	18 (60)	9 (39)	9 (41)	0.90
**Vaccination status, *n* (** **%** **)**				
ChAdOxl-1 dose		16 (69)		
ChAdOx1-2 dose		2 (9)		
CoronaVac-2 dose		5 (22)		
Interval after last vaccine (d), median (IQR)		49 (21–62)		

eGFR, estimated glomerular filtration rate; mTOR, mammalian target of rapamycin.

**Table 2 vaccines-10-00572-t002:** Hospital course of COVID-19 infection in kidney transplant recipients.

Clinical Course, *n* (%)	Total (N = 45)	Vaccinated (N = 23)	Unvaccinated (N = 22)	*p*-Value
**Clinical manifestation**				
Asymptomatic	2 (5)	1 (4)	1 (5)	0.95
Fever	41 (93)	22 (96)	19 (90)	0.50
Upper respiratory tract symptoms	15 (34)	9 (39)	6 (29)	0.46
Pneumonia	31 (70)	16 (70)	15 (71)	0.89
Diarrhea	2 (5)	1 (4)	1 (5)	0.95
**Treatment**				
Favipiravir	41 (91)	21 (91)	20 (91)	0.96
Remdesivir	2 (4)	0 (0)	2 (9)	0.14
Dexamethasone	17 (38)	11 (48)	6 (27)	0.16
Methylprednisolone	3 (7)	1 (4)	2 (9)	0.52
Tocilizumab	1 (2)	1 (4)	0 (0)	0.32
Baricitinib	2 (4)	0 (0)	2 (9)	0.32
Hemoperfusion	1 (2)	1 (4)	0 (0)	0.32
**Oxygen therapy**				0.55
No requirement	17 (39)	11 (48)	6 (29)	
Nasal cannula	15 (34)	7 (30)	8 (38)	
High flow nasal cannula	6 (14)	3 (13)	3 (14)	
Invasive mechanical ventilation	6 (14)	2 (9)	4 (19)	
**Immunosuppressive management**				
**Calcineurin inhibitors (N = 41)**				0.45
Continue	34 (83)	18 (82)	16 (84)	
Reduce dose	4 (10)	3 (14)	1 (5)	
Discontinue	3 (7)	1 (4)	2 (11)	
**Mycophenolate (N = 42)**				0.47
Continue	12 (29)	7 (30)	5 (26)	
Discontinue	30 (71)	16 (70)	14 (74)	
**mTOR inhibitors (N = 3)**				0.61
Continue	3 (100)	1 (100)	2 (100)	

mTOR, mammalian target of rapamycin.

**Table 3 vaccines-10-00572-t003:** Outcomes of COVID-19 infection in kidney transplant recipients classified by vaccination status.

Outcome, *n* (%)	Total		Vaccinated		Unvaccinated	RR ^†^ (95% CI)	*p*-Value
	(N = 45)	Any Vaccine (N = 23)	Full Dose ^‡^ (N = 7)	AZ 1 Dose (N = 16)	(N = 22)		
Death	11 (24)	3 (13)	0 (0)	3 (19)	8 (36)	0.56 (0.29–0.83)	0.03
Oxygen requirement	28 (62)	11 (50)	3 (43)	9 (56)	17 (74)	0.35 (0.1–1.23)	0.10
Mechanical ventilation	6 (14)	2 (9)	0 (0)	2 (13)	4 (19)	0.57 (0.12–2.73)	0.48
Bacterial pneumonia	14 (31)	6 (26)	1 (14)	5 (31)	8 (36)	0.7 (0.2–2.51)	0.59
IPA	5 (11)	1 (4)	0 (0)	1 (6)	4 (18)	0.23 (0.02–2.21)	0.20
Pulmonary embolism	4 (9)	3 (13)	0 (0)	3 (19)	1 (5)	1.5 (0.41–6.93)	0.69
Septicemia	4 (9)	2 (9)	1 (14)	1 (6)	2 (9)	1.05 (0.14–8.18)	0.96
Acute kidney injury	21 (47)	11 (50)	1 (14)	10 (63)	10 (46)	0.76 (0.23–2.46)	0.64
AKIN stage 1	12 (27)	7 (30)	1 (14)	6 (38)	5 (23)	1.06 (0.28–3.98)	0.93
AKIN stage 2	5 (11)	1 (4)	0 (0)	1 (6)	4 (18)	0.23 (0.03–2.21)	0.20
AKIN stage 3	3 (7)	2 (9)	0 (0)	2 (13)	1 (5)	2.2 (0.19–16.12)	0.53
Required KRT	2 (4)	1 (4)	0 (0)	1 (6)	1 (5)	0.98 (0.23–4.76)	0.58

AKIN, Acute Kidney Injury Network; AZ, ChAdOx1 vaccine (Oxford-AstraZeneca); CI, confidence interval; IPA, invasive pulmonary aspergillosis; KRT, kidney replacement therapy; RR, relative risk ratio. ^†^ statistically compared between vaccinated and unvaccinated group. **^‡^** CoronaVac, *n* = 5; ChAdOx1, *n* = 2.

**Table 4 vaccines-10-00572-t004:** Univariate and multivariate analysis of factors associated mortality in COVID-19 infected kidney transplant recipients.

Factors	Univariate Analysis		Multivariate Analysis	
	OR (95% CI)	*p*-Value	OR (95% CI)	*p*-Value
Vaccine	0.56 (0.29–0.83)	0.03	0.54 (0.24–0.83)	0.03
Recipient age	1.01 (0.96–1.07)	0.56		
Deceased donor transplantation	1.46 (0.23–6.53)	0.63		
Obesity	2.20 (0.55–8.74)	0.26		
Diabetes	4.89 (1.09–21.95)	0.04	5.37 (1.07–26.95)	0.04
Sepsis	3.56 (0.44–28.89)	0.24		

CI, confidence interval; OR, odds ratio.

## Data Availability

Data are available upon request.

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
