# Peer review of "Benefits of Inactivated Vaccine and Viral Vector Vaccine Immunization on COVID-19 Infection in Kidney Transplant Recipients"

_vaccines, 2022, doi:10.3390/vaccines10040572_

Round 1

Reviewer 1 Report

I think the research question is a good one, the issues I had identified is the very low sample size that avoids making fully conclusions and outcomes interpretations. I'd recommend the authors to try to balance the two cohorts, the fact that only the age seems statistical different without any intention to balance, could generate bias on the results. In addition, the rest of cofounders may not show statistically different due the very small sample size, so there could be certain possibility still that the analysis could be bias. Just to double-check there are none of these problems within the comparison, probably the inclusion of a negative control outcome will be of utility. This would be strongly recommended to the authors, in case there is no possibility to extend the database. Lastly, in the discussion section should be an extended discussion on what would be the risks (not only in mortality) for those vaccinating with one covid-19 and those fully vaccinated (two shots), sample size may not help but it deserves a good try on the discussion, we don't wish to give the impression that vaccinating with one dose is sufficient (it may be interesting to include as well to include the time since the subject received their last vaccine dose up to when they were confirmed with Covid-19, just to explore persistence topics). The introduction could also be improved in the sense that more vaccine effectiveness and relative risk values could be offer for those kidney transplant receipts that were infected with covid-19, just to provide the right context for these outcomes, this will be of benefit. Overall, the article looks good but definitely it needs to mitigate the current high risk of bias.

Author Response

Response to reviewer(s)’ comments:

Reviewer 1

I think the research question is a good one, the issues I had identified is the very low sample size that avoids making fully conclusions and outcomes interpretations. I'd recommend the authors to try to balance the two cohorts, the fact that only the age seems statistical different without any intention to balance, could generate bias on the results. In addition, the rest of cofounders may not show statistically different due the very small sample size, so there could be certain possibility still that the analysis could be bias. Just to double-check there are none of these problems within the comparison, probably the inclusion of a negative control outcome will be of utility. This would be strongly recommended to the authors, in case there is no possibility to extend the database.

Response: Vaccinated group in our study was older than unvaccinated group but has lower mortality rate. We have added in DISCUSSION part to point this out (line 207-210). We have performed multivariate analysis (table 4) to check for confounding factors including patients’ age to minimize risk of bias and found that vaccination was the only significant factor associated with lower mortality. Unfortunately, we did not have negative control because we aim to study the efficacy of vaccination in reducing mortality not in preventing infection.

Lastly, in the discussion section should be an extended discussion on what would be the risks (not only in mortality) for those vaccinating with one covid-19 and those fully vaccinated (two shots), sample size may not help but it deserves a good try on the discussion, we don't wish to give the impression that vaccinating with one dose is sufficient (it may be interesting to include as well to include the time since the subject received their last vaccine dose up to when they were confirmed with Covid-19, just to explore persistence topics).

Response: We have added in the discussion and emphasize that 2 doses of vaccination is much better one dose of vaccine. The time since the subject received their last vaccine dose up to when they were confirmed with Covid-19 was also added in the RESULT part (line 153-154). From this data, we also discussed about shortening the interval between first and second dose of viral vector vaccine in the DISCUSSION (line 211-222).  

The introduction could also be improved in the sense that more vaccine effectiveness and relative risk values could be offer for those kidney transplant receipts that were infected with covid-19, just to provide the right context for these outcomes, this will be of benefit. Overall, the article looks good but definitely it needs to mitigate the current high risk of bias.

Response: We have added more information in the introduction part to make it more informative. (line 54-66)

Reviewer 2 Report

The manuscript (vaccines-1629106) entitled “Benefits of inactivated vaccine and viral vector vaccine immunization on COVID-19 infection in kidney transplant recipients” by Dr. Sutharattanapong and colleagues is a retrospective study which included a total of 45 kidney transplant recipients kidney transplant recipients. The main results indicate that inactivated and viral vector COVID-19 vaccines have beneficial effect on mortality reduction in kidney transplant recipients. Despite the introduction should be improved, the ms is in general concise, well written and well organized. The work will increase our knowledge on COVID-19 vaccination and kidney transplant clinical outcome. As stated by the authors, the main limitation of the study is the small sample size. In my opinion, the manuscript can be accepted following a major revision. I have several observations for improving the manuscript: 

Major comments
1.    The introductive section should be enlarged pointing on study background informations. For instance, the authors can more deeply describe the status of both humoral immune and kidney transplantation Thailand. A brief overview of the currently employed vaccines should be included as well.  A large variety of reports have been published in these fields. The authors can check  PMID: 35308704, PMID: 35270775, PMID: 34669595 and PMID: 33902459, among others. Other important references on sars-cov-2 and kidney transplantation  are as follows PMID: 35304094, PMID: 35301103, PMID: 35283457 and PMID: 35281011.
2.    I suggest including at least patients ages in the “Study design and participants” section 
3.    As stated by the authors, the main limitation of the study is the small sample size.
4.    Figures and tables are demonstrative 

Minor observations
Line 46 the efficacy and safety of vaccination in preventing COVID-19 infection has also been described in detail here (DOI: 10.3390/v13091687). This reference should be included
Lines 58-72 supporting references should be included
Lines 96-107 supporting references should be included
Lines 167-170 I suggest rephrasing the sentence in order to remove pharentesis
Line 175 the name of the institute should be included
Line 186 it is unclear the comparison between the incidence of pneumonia in kidney transplant recipients with that obtained by Pongpirul et al in the general population
Line 197 Data on humoral immune of kidney transplant recipients following COVID-19 vaccination are reported here (DOI: 10.3390/vaccines10030385). This reference should be included, while the work discusses 

Author Response

Response to reviewer(s)’ comments:

Reviewer 2

Major comments
1.    The introductive section should be enlarged pointing on study background informations. For instance, the authors can more deeply describe the status of both humoral immune and kidney transplantation Thailand. A brief overview of the currently employed vaccines should be included as well.  A large variety of reports have been published in these fields. The authors can check  PMID: 35308704, PMID: 35270775, PMID: 34669595 and PMID: 33902459, among others. Other important references on sars-cov-2 and kidney transplantation  are as follows PMID: 35304094, PMID: 35301103, PMID: 35283457 and PMID: 35281011.

Response: We have added the information and importance of vaccination in inducing humoral and cell-mediated immune response to prevent severe disease and death from Covid. Varied immune response to different kinds of vaccine was also mentioned. Some of the suggested references were also included. (line 48-66)

  1.    I suggest including at least patients ages in the “Study design and participants” section 

Response: We have rephrased and added patients’ age in the section. (line 73-75)

  1.    As stated by the authors, the main limitation of the study is the small sample size.
    4.    Figures and tables are demonstrative 

Minor observations
Line 46 the efficacy and safety of vaccination in preventing COVID-19 infection has also been described in detail here (DOI: 10.3390/v13091687). This reference should be included

Response: Reference was included. (line 48, reference number 4)

Lines 58-72 supporting references should be included

Response: References were included. (line 83-84, reference number 20 and 21)

Lines 96-107 supporting references should be included

Response: Reference was included. (line 135, reference number 23 and line 138, reference number 24)

Lines 167-170 I suggest rephrasing the sentence in order to remove pharentesis

Response: We have rephrased the sentence and remove parentheses. (line 211-216)

Line 175 the name of the institute should be included

Response: We have included the name of our institution. (line 227)

Line 186 it is unclear the comparison between the incidence of pneumonia in kidney transplant recipients with that obtained by Pongpirul et al in the general population

Response: We would like to show that kidney transplant recipients have more severe disease and much higher incidence of pneumonia than general population to emphasize the need for vaccination.

Line 197 Data on humoral immune of kidney transplant recipients following COVID-19 vaccination are reported here (DOI: 10.3390/vaccines10030385). This reference should be included, while the work discusses 

Response: This article was published after we had submitted this manuscript. The reference was added. (line 259, reference number 33)

Round 2

Reviewer 1 Report

I'm ok with the responses provided. Thanks so much

Reviewer 2 Report

The ms can be accepted for publication